# Objectively-Measured Physical Activity and Sedentary Behaviors and Related Factors in Chinese Immigrants in the US with Prior Gestational Diabetes Mellitus

**DOI:** 10.3390/ijerph191811409

**Published:** 2022-09-10

**Authors:** Shuyuan Huang, Garrett I. Ash, Soohyun Nam, Sangchoon Jeon, Erin McMahon, Robin Whittemore

**Affiliations:** 1Rutgers-NYU Center for Asian Health Promotion and Equity, NYU Rory Meyers College of Nursing, New York, NY 10010, USA; 2Pain Research, Informatics, Multi-Morbidities, and Education (PRIME), VA Connecticut Healthcare System, West Haven, CT 06516, USA; 3Yale School of Medicine, Yale University, New Haven, CT 06510, USA; 4Yale School of Nursing, Yale University, West Haven, CT 06477, USA; 5College of Nursing, The University of Arizona, Tucson, AZ 85721, USA

**Keywords:** physical activity, sedentary behaviors, Chinese immigrants, gestational diabetes, diabetes prevention, women’s health, acculturative stress, social support

## Abstract

Chinese immigrants in the US are disproportionately affected by gestational diabetes mellitus (GDM) and type 2 diabetes (T2D). The aims of this study were to describe their physical activity (PA) and sedentary behaviors (SB) patterns and to identify determinates of objectively-measured PA and SB among Chinese immigrants in the US with prior GDM. We conducted a cross-sectional study among 106 Chinese immigrants with prior GDM across the US. PA and SB were measured by GT9X+ hip accelerometers for 7 consecutive days. Validated questionnaires in English and Chinese were used to assess knowledge and risk perceptions as well as cultural and psychosocial characteristics. Descriptive, bivariate, and multiple regression analyses were performed. Only 27% of participants met the PA guidelines. The median duration of moderate–vigorous-intensity PA (MVPA) per week was 79 (IQR 38–151) minutes. Participants had an average of 9.2 ± 1.4 h of sedentary time per day. Living with parents (who may provide family support) was associated with more MVPA minutes per week, more steps per day, and a greater likelihood of meeting PA guidelines. Higher levels of acculturative stress were associated with fewer MVPA minutes per week. Being employed and having a lower BMI were associated with more SB. Strategies are needed to increase MVPA among this high-risk group, including decreasing acculturative stress and increasing family support. Different strategies are needed to decrease SB among this population.

## 1. Background

People with prior gestational diabetes mellitus (GDM) have ten times the risk of developing type 2 diabetes (T2D) compared to people without prior GDM [1]. Hence, it is important to prevent or delay the development of T2D in this high-risk group. In 2020, the estimated prevalence of GDM was 7.8% in the US, while it was 13.5% for Chinese people in the US [2]. In addition, immigrants had a higher prevalence of GDM [3,4] and a higher risk for T2D [5] compared to their non-immigrant counterparts. Chinese immigrants represent one of the fastest growing and largest foreign-born groups (three million) [6] in the US with a unique language, culture, genetic background, physical activity, and other lifestyle behaviors that contribute to GDM [7]. Lack of culturally sensitive care is often associated with poorer self-management and glycemic control [8], and a higher risk of end-stage renal disease among Chinese immigrants with T2D [9]. Hence, it is important to have culturally sensitive care to prevent the risk of T2D among Chinese immigrants with prior GDM.

Physical activity (PA) and sedentary behaviors (SB) are modifiable behaviors that reduce the risk for T2D. Large trials and systematic reviews have shown that lifestyle interventions targeting PA in the postpartum period can be effective in reducing the incidence of T2D among people with prior GDM, increasing insulin sensitivity, or decreasing weight [10,11]. Although SB is less studied among people with prior GDM, reducing SB has been shown to be beneficial for glucose regulation and can contribute to better T2D management in the general population [12]. Both the total amount and diurnal pattern of SB affect risk levels. Prolonged SB time is one of the patterns of SB that independently affects cardio-metabolic biomarkers and glucose and insulin sensitivity [13]. In other words, interrupting SB, such as prolonged sitting, every 30 min can improve postprandial glucose, insulin, lipid profiles [14], and all-cause mortality [15]. 

In the few studies that had been conducted among Chinese immigrants in the US, insufficient PA has been reported [16,17], especially with a low level of moderate- to vigorous-intensity physical activity (MVPA) or recreational PA [16]. However, these studies have primarily reached older male and female Chinese immigrants in New York City with limited generalizability to a younger generation of reproductive-age females who migrated to the US in recent decades. Identifying the unique risk factors of young Chinese immigrant mothers with prior GDM may contribute to intervention targets. Furthermore, none of the studies used objective measures for PA, which would be more accurate and address challenges of recall bias and social desirability issues from self-report measures. 

Knowledge, risk perception, psychosocial, and cultural factors are likely to influence the PA and SB of Chinese immigrants with prior GDM. Specifically, lack of PA knowledge, low perceived risk for T2D, low social support, or low PA self-efficacy can contribute to suboptimal PA among people with prior GDM [18]. In addition, there are unique factors associated with low PA levels among the immigrant population. Lower levels of acculturation to the US culture were associated with low PA levels in a group of older Chinese immigrants [17]. Acculturative stress is caused by the stressors during the acculturation process including social isolation, language barriers, and discrimination [5,19], which can influence health behaviors, such as PA and SB. In addition, there are Chinese cultural practices that limit PA and promote SB during the postpartum period. The postpartum practice of “doing the month” confines people to their homes for one month after birth to facilitate postpartum recovery and infant care [20]. Alternatively, another Chinese cultural practice of living with extended family (e.g., parents and/or in-laws of the mothers) might allow Chinese immigrants to receive considerable support for childcare and housework [21]. This parental support may enable them to implement more self-care activities, including being more physically active.

The aims of this study were: (1) to describe the patterns of objectively-measured PA and SB among Chinese immigrants in the US with prior GDM, and (2) to identify PA knowledge, T2D risk perception, cultural (acculturation, living with parents/in-laws, and postpartum practices), and psychosocial factors (acculturative stress, PA social support, and PA self-efficacy) associated with PA and SB after controlling for covariates. 

## 2. Methods

### 2.1. Study Design, Settings, and Sampling

We conducted a cross-sectional study between August 2020 and August 2021 among Chinese immigrants with prior GDM across the US. We recruited participants via multiple methods, using social media, snowball sampling, flyers, and a study website. Flyers were also sent to the Maternal-Fetal Medicine clinic in New Haven, CT, Queens Public Libraries in New York City, the Chinese Nail Salon Association, and New York Chinese Doctors Association. Social media platforms were the primary recruitment methods, including WeChat (a popular Chinese social media platform) and Facebook. We enrolled a nationwide convenience sample of 106 Chinese immigrants with prior GDM. Our inclusion criteria were: (1) age between 18 to 45 years; (2) self-identified as Chinese; (3) born outside of the US; (4) residing in the US; (5) self-reported diagnosis of GDM during at least one pregnancy; (6) speak English or Chinese (Mandarin); (7) at least one child delivered from the index pregnancy was between 6 months to 5 years old. The exclusion criteria were: (1) diagnosis of type 1 diabetes (T1D) or T2D; (2) conditions that interfere with PA (e.g., heart failure or injury); (3) currently pregnant.

Institutional Review Board approval (HIC: #2000027360) from Yale University was obtained for this study before data collection. Participants who were interested in the study contacted the principal investigator via their preferred social media. Participants were provided a brief description of the study, and if interested they were screened for eligibility using a 10-question Qualtrics^TM^ survey. Informed consent was obtained from all participants. We collected contact information from enrolled participants and mailed the GT9X+ ActiGraph to them, along with the instructions, a daily diary for logging wear time (both in English and in Chinese), and a prepaid return envelope. Participants were asked to wear the hip-worn accelerometer for 7 consecutive days during all waking time (removal for sleep and shower/bath). We sent a daily text/email reminder via Qualtrics every morning to promote adherence to the study protocol. At the end of the 7-day wear-time, participants completed an online Qualtrics survey of study questionnaires. Participants received a $10.00 gift card after their devices were returned. We also had a lottery to provide two $50.00 gift cards among all participants who completed the online Qualtrics survey at the end of the study. 

### 2.2. Measurements and Variables

#### 2.2.1. Covariates 

*Socio-demographic data* included age, education, employment status, annual household income, marital status, birthplace, age at the time of immigration to the US, immigration status, length of stay in the US, and preference for language use. *Health-related data* included chronic illness comorbidities, family history of diabetes, number of children, number of pregnancies with GDM, time since last birth, GDM treatment methods during pregnancy (i.e., insulin), history of postpartum depression, and self-reported height and weight for body mass index (BMI, kg/m^2^).

#### 2.2.2. PA & SB

*PA and SB* were measured objectively with a hip-worn ActiGraph GT9X+ accelerometer (ActiGraph, Pensacola, FL, USA) for 7 consecutive days. We manually validated the wear-time in the ActiLife^TM^ software program along with participants’ wear-time diary. We defined non-wear time as ≥60 consecutive minutes of zero activity intensity counts, with allowance for 1–2 min of counts between 0 and 100 [22]. Forty-three days (5.8%) were excluded due to <10 h valid wear-time, and 3 (2.8%) participants were excluded due to <4 valid days or <1 valid weekend day [22]. We used the ActiLife^TM^ software program to generate minutes per week in light, moderate, and vigorous-intensity activity, and average daily steps. We used a well-established standard [22] for PA intensity cut-points (light 100–2019 Count/min, moderate 2020–5998 Count/min, and vigorous 5999 Count/min) which has also been used among postpartum people [23]. Minutes of MVPA per week were calculated, and participants were categorized into meeting PA guidelines (≥150 min of MVPA per week) or not. We also counted the number of days with ≥30 min MVPA and the number of days with <30 min MVPA when the previous day was also MVPA < 30 min (i.e., consecutive inactive days) according to the ADA 2021 Standard of Care on PA [24]. Lastly, any activity with 0–99 count/min was categorized as sedentary. Prolonged sedentary time (sedentary time accumulated in bouts of ≥30 min) and non-prolonged sedentary time (sedentary time accumulated in bouts <30 min) were calculated separately. 

#### 2.2.3. Risk Perception & Knowledge

*T2D Risk Perception* was measured using the 12-item Chinese Risk Perception Survey for Developing Diabetes (RPS-DD) [25]. We assessed the overall perceived risk (1 item, range 1–4) as well as Diabetes Risk knowledge (11 items, range 0–1). A greater score indicates higher perceived risk and more knowledge of diabetes risk, respectively. PA knowledge or awareness of the PA guidelines was assessed by one question: “Do you know how many minutes of moderate- to vigorous-intensity physical activity is recommended for your age group in the national guidelines?” 

#### 2.2.4. Cultural Factors 

*Acculturation* was measured using the Stephenson Multigroup Acculturation Scale (SMAS), a 32-item scale with two dimensions, ethnic society immersion (first 17 items, score range 1–4) and dominant society immersion (items 18–32, score range 1–4) [26]. The degree of immersion is measured at an individual level by evaluating different domains of life, including language use, ethnicity of friends, food, and media preference. Ethnic society immersion is a measure of how closely the individual identifies with his or her heritage culture. Dominant society immersion is a measure of how much the individual is immersed in the host culture, namely the culture of the US in this study. The internal consistency coefficients (Cronbach’s alpha) were 0.86 for both subscales in this study. 

*Living with parents/in-laws.* Participants were asked about the length of any parent/in-law’s stay in the US with them since the birth of their child (current or past), a common practice in many Chinese immigrant families, usually to help with childcare and household activities during pregnancy and the postpartum period [21]. 

*Postpartum practices*. We adapted the 27-item Adherence to Doing-the-Month Practices (ADP) scale to evaluate participants’ current degree of acceptance for the Chinese postpartum practices (range 1–5) [27]. The items of ADP are about the traditional Chinese beliefs on the diet, PA, and other lifestyle behaviors in the first month postpartum. A higher ADP score indicates better acceptance of Chinese postpartum practices. The Cronbach’s alpha was 0.90 for the ADP scale in this study. 

#### 2.2.5. Psychosocial Factors 

*Acculturative stress* was assessed using the Migration–Acculturation Stressor Scale (MASS), a 22-item scale assessing the degree of stress from various acculturation-related challenges, including homesickness, the cultural difference between home culture and the US culture, social isolation, and unfamiliar climate (range 1–5) [19]. A higher MASS score indicates higher acculturative stress. The Cronbach’s alpha for the scale was 0.91 in this study. 

*Social Support for PA (SSPA)* has 6 items, three for family and three for friends [28], with a higher score (range 0–4) indicating higher social support for PA. A Chinese version of SSPA is available [29]. The Cronbach’s alpha was 0.82 for the family subscale and 0.84 for the friend subscale in this study. *Self-Efficacy for PA (SEPA)* has four items [27], with a higher score (range 1–4) indicating higher self-efficacy for PA. A Chinese version of SEPA is available, too [29]. The Cronbach’s alpha for the SEPA scale was 0.79 in this study.

### 2.3. Sample Size and Data Analysis

The sample size of this study was determined based on the effect size of a previous study on psychosocial correlates of MVPA [30]. In power analysis, 100 participants would be sufficient to have a power of 90% at a 5% significance level (2-sided) to detect a medium correlation of 0.32, which is equivalent to an R-square of 0.10 in regression models. This sample size had acceptable power of >80% to detect the partial correlation of 0.28 in multiple regression with covariates. 

All data management and statistical analysis were conducted using SAS^TM^ v. 9.4. Descriptive analyses were performed to describe demographic and clinical characteristics, key predictors, and distributions of outcome variables. Since the distribution of MVPA minutes per week was highly skewed, we log-transformed it for further analysis. Bivariate associations between the covariates, the other predictors (risk perception, knowledge, cultural factors, and psychosocial factors), and the continuous outcome variables (weekly MVPA, daily steps, prolonged SB time per day, and percentage of SB time) were assessed using Pearson, Spearman, or point-biserial correlation coefficients. 

Primary outcomes including MVPA minutes per week, meeting PA guidelines, daily steps, prolonged SB time, and percentage of SB time were examined using the generalized linear model (GLM) and logistic regression model (LRM). We conducted unadjusted GLM and LRM to select candidates of predictors and covariates with a *p*-value of <0.10. We developed adjusted models to examine the effects of predictors after controlling for selected covariates. Parsimonious models were developed with a hierarchical regression model approach (See the five steps listed below). We assessed multicollinearity among covariates and residuals for outlier and normality assumptions. 

Model 1: covariates (Base model); 

Model 2: covariates + risk perception & knowledge (to select significant factors); 

Model 3: covariates + cultural factors (to select significant cultural factors); 

Model 4: covariates + psychosocial factors (to select significant psychosocial factors);

Model 5: covariates + risk perception & knowledge + cultural factors + psychosocial factors (factors selected from those variables significant at 0.1 level in model 2 & 3 & 4). 

## 3. Results

### 3.1. Description of Sample Characteristics and Key Variables

All 106 participants were recruited via social media. The mean age of the participants was 34.3 ± 3.7 years, and a mean BMI of 21.7 ± 2.6 kg/m^2^ (see Table 1). The mean age of the arrival in the US was 25.0 ± 5.5 years, while the mean length of the US stay was 9.2 ± 4.8 years. The majority of participants were highly educated (72.1% had a master’s degree or higher), employed (67.3%), had an annual household ≥ $80 K (69.9%), from mainland China (96.2%), were US citizens (15.4%) or permanent residents (51.9%), and married with 1–2 children (92.5%). Approximately half of the participants had given birth within the past 6–12 months. The majority of the participants had one pregnancy with GDM (73.6%), did not use insulin as GDM treatment (83.0%), had no family history of diabetes (67.9%), currently were not breastfeeding (63.2%), and had no history of postpartum depression (94.3%). About 35% (n = 37) of participants chose the Chinese language for the online Qualtrics survey. A description of the key predictors in the study is reported in Table 2. 

Participants wore the accelerometers for 7 (IQR 7–7) days with an average length of 14.2 ± 1.4 h per day (night sleep time was not measured) (see Table 3). The median duration of MVPA per week was 79 (IQR 38–151) minutes. About 27% (n = 28) of the participants met the PA guidelines of 150 min or more of MVPA per week. Over half of the participants did not have 30 min of MVPA on any day of the week. The participants spent on average 4.2 ± 2.2 days (range 1–6 days) with MVPA of less than 30 min when the previous day was also with less than 30 min of MVPA. There was very little vigorous PA time (0 (IQR 0–0.29) minutes) per week among the participants. The average daily steps were 5908 ± 2138 steps. Participants had an average of 9.2 ± 1.4 h of SB time per day, which was about 64.7 ± 7.8% of the total wear time. The average prolonged SB time was 2.5 ± 1.4 h per day, while the average non-prolonged SB time was 6.7 ± 1.1 h. 

### 3.2. Determinants of PA

In the bivariate analysis, four covariates (higher BMI, education with a master’s degree or higher, preference for English, and annual household income more than $80 K) were positively associated with participants’ MVPA minutes per week (*p* < 0.10). In addition, participants with no knowledge of PA guidelines, lower knowledge of T2D risk, those currently not living with parents, with higher acceptance of the postpartum practice, higher acculturation stress, and lower social support for PA had fewer minutes of MVPA per week (*p* < 0.10). In the final adjusted GLM (Table 4), currently not living with parents (−0.48 (SE = 0.23), *p* = 0.041), and having higher acculturative stress (−0.47 (SE = 0.14), *p* = 0.001) were significantly associated with fewer MVPA minutes per week after controlling for covariates. Participants’ BMI, household income, knowledge of PA guidelines, and PA social support also remained in the final model, even though not statistically significant at a 0.05 significance level. Moreover, PA social support was significantly correlated with currently living with parents (spearman r = 0.20, *p* = 0.044). The R-square value for the final model was 0.31. 

In the final adjusted LRM for meeting PA guidelines (Table 5), participants who were permanent residents (OR = 0.155, 95% CI 0.036–0.664), with the age of youngest child ≥ 1 (OR = 0.244, 95% CI 0.079–0.755), and not living with their parents (OR = 0.180, 95% CI 0.054–0.602) were less likely to meet the PA guidelines after controlling for covariates. The Hosmer–Lemeshow goodness-of-fit test demonstrated a good model fit (*p* = 0.659). Similarly, in the final GLM for steps per day (Table 6), participants who were not living with their parents (β = −1682 ± 523, *p* = 0.002) had fewer steps per day. The R-square value for this final model was 0.09. 

### 3.3. Determinants of SB

In the bivariate analysis, participants who were older and had an older age of arrival in the US had lower prolonged SB time per day, while participants who had higher education (master’s degree or higher), were employed, with an annual household income ≥ $80 K, and with just one child had longer prolonged SB time per day (*p* < 0.10). In the final adjusted GLM (Table 7), participants who were employed (β = 1.21 ± 0.26, *p* < 0.001) had significantly longer prolonged SB time. The only other variable that remained in the final model was the number of children, with participants who had just one child having a greater prolonged SB time, which was not statistically significant at a 0.05 level. The R-square value for this final model was 0.22. 

In addition, we also explored the predictors for the percentage of SB time. In bivariate analyses, participants’ BMI was negatively associated with the percentage of SB time, while participants with a master’s degree or higher, being employed, with just one child, and those with higher PA self-efficacy had a higher percentage of SB time (*p* < 0.10). In the final adjusted GLM (Table 8), participants who had lower BMI (β = −0.007 ± 0.003, *p* = 0.006) and those who were employed (β = 0.077 ± 0.014, *p* < 0.001) had a significantly higher percentage of SB time. The R-square value for this final model was 0.29. 

## 4. Discussion

Chinese immigrants with prior GDM have an inactive and sedentary lifestyle pattern as identified in this study, using an objective measure of PA and SB. All three PA-related outcome variables were associated with currently living with parents. Weekly MVPA was also significantly negatively associated with acculturative stress. Less weekly MVPA was associated with no knowledge of PA guidelines and having lower social support for PA, but the relationships were not statistically significant. Not meeting PA guidelines was significantly associated with being permanent residents (instead of US citizens) and with the age of the youngest child ≥ 1. Knowledge and risk perception as well as cultural and psychosocial factors were not associated with SB. However, being employed was associated with both prolonged and total SB time, while lower BMI was associated with a higher total SB time. 

The PA level in our participants was low. Only 27% of Chinese immigrants met the PA guidelines of ≥150 min of MVPA per week. Most participants had little to no vigorous activities, and more than half of the participants did not have 30 min of MVPA on any day during the week. Thus, over 70% of participants who did not meet PA guidelines in this study, which was higher than the percentage (55–64%) among general adults in the US who do not meet the PA guidelines for aerobic exercise [31]. Also, participants in this study spent only about 1.28% (IQR 0.59–2.49%) of the total wear time in MVPA, which was less than that of overweight/obese adults (2.2 ± 2.1%) with T2D in one study [32]. In addition, the average steps per day among the participants (5908 ± 2138 steps) were much lower than the average steps per day (9209 ± 3254steps) in another group of postpartum people [23]. In general, the low PA level among our participants was consistent with the suboptimal PA level reported among other populations with prior GDM [33] and other older Chinese immigrants in the US [16].

SB in this sample of Chinese immigrants with previous GDM was high. Both the SB hours per day (9.2 ± 1.4 h) and the percentage of SB time (64.7 ± 7.8%) in this sample were higher than that of non-pregnant general populations in the US who had less than 8 h of SB time per day and less than 60% of total wear time being sedentary [34]. However, the percentage of SB time (64.7 ± 7.8%) and the prolonged SB time (2.5 ± 1.4 h) in this study were less than that for a group of adults older than 45 years old (77.4 ± 9.4% of SB time, while about 48% SB were prolonged SB time) [15]. This difference is not surprising as the average SB time usually increases by age among adults [35]. Nevertheless, in clinical practice, education on both PA and SB for people with prior GDM is needed. In addition, referrals to existing programs and resources for diabetes prevention, like the National Diabetes Prevention Program are recommended. The development of a culturally sensitive National Diabetes Prevention Program that is tailored to the unique needs of immigrants with recent GDM is needed as well.

Based on our findings, acculturative stress is a potentially modifiable factor for increasing MVPA minutes per week but not for decreasing SB among Chinese immigrants with prior GDM. Few studies have examined the relationship between acculturative stress and PA. As previously reported in a systematic review of 168 studies published between the years 1984 and 2012, the association between psychological stress and PA was mixed [36]. Unfortunately, none of the studies evaluated acculturative stress. In a more recent study among Korean immigrants in the US, acculturative stress was associated with SB time but not PA level [37]. In an intervention study conducted in Korea among Korean-Chinese migrants, a 24-week walking program reduced participants’ acculturative stress levels [38], suggesting acculturative stress can be mitigated by more PA. In summary, more studies are needed to verify the relationship between acculturative stress and PA and SB in different immigrant populations; however, assessing and addressing acculturative stress is important in Chinese immigrants with prior GDM. Future intervention could consider the inclusion of stress reduction strategies to promote a healthy active lifestyle, especially among the immigrant population who experience unique acculturative stressors. 

Living with parents was a protective factor for being more physically active (e.g., more MVPA minutes per day, more steps per day, and a higher likelihood of meeting PA guidelines) but not for decreasing SB among this sample of Chinese immigrants with prior GDM. It is common for Chinese immigrant families to be multi-generational households [39]. The parents of Chinese immigrants often come to the US to reunite with family, and they provide care to the children and assist with household work [21,39]. The important role of the Chinese immigrants’ parents in the family was partly demonstrated by the positive association between social support and living with parents in this study. Parents in multi-generational households can help new families by providing childcare, housework, emotional and parenting support, as well as social support to PA. In immigrant families without extended family support, interventions are needed to support new families in promoting healthy lifestyles. 

Several covariates that were significantly associated with PA and/or SB can help identify high-risk groups for low PA and high SB. First, Chinese immigrants who were permanent residents were less active compared to those who were US citizens. Permanent residents can be very different from those who chose to become naturalized citizens of the US; the latter have more identification with US culture and had more resources or extended family and support in the US [40]. It was also interesting to note that the Chinese immigrants with children < 1 year old were more likely to meet PA guidelines. This could be due to their recent experience with their GDM diagnosis during pregnancy which may have motivated them to maintain a healthy lifestyle within one year postpartum [41]. Capitalizing on this teachable moment and engaging people with prior GDM in earlier postpartum time (or even starting during pregnancy) is warranted to restore or promote a healthy lifestyle [42]. However, there is limited high-quality research focusing on postpartum PA intervention among people with prior GDM [43]. More cost-efficient interventions with sufficient length (longer than one year postpartum) are needed as well as the maintenance of an active lifestyle. Lastly, there were unique factors affecting PA and SB, indicating that different strategies are needed to improve the two lifestyle behaviors, respectively. Decreasing employment-related SB time may be needed for these young working immigrant mothers. For instance, workplace policies are needed to encourage moving throughout the day. 

There are some limitations of this study. First, this is a cross-sectional study so we cannot make any causal inference. The study was conducted during the COVID-19 pandemic and participants’ PA and SB may have been influenced by the quarantine or stay-at-home policy at the time of data collection. The pandemic and vaccination timing also matters. As people’s perception of COVID-19 risks and severity evolves, their lifestyle behaviors may change accordingly. In addition, the sample in this study was highly educated with high income. However, it is similar to the population characteristics of Chinese immigrant population in this age group based on the American Community Survey data by US Census Bureau [6]. Chinese immigrants are more likely to have an advanced degree than immigrants overall and US-born adults, as their primary reason for immigration is to pursue higher education [44]. This leads to higher average income and employment in science, engineering, information technology, management, and business professions [45]. This high education and income level can affect their knowledge of PA and T2D risk, as well as the PA resources available to them. More research is needed on Chinese immigrants across different income strata. In addition, there may be sampling bias as the participants were all recruited via social media. However, the use of a mobile phone (over 90%) and WeChat (over 70%) is highly prevalent among a group of low-income older Chinese immigrants [46], who have a need for social support and social connection with people from their home country. We suspect that mobile phones and WeChat are even more prevalent among a more educated group of middle-aged Chinese immigrants. Lastly, the BMI covariate was based on self-reported measures that could be biased due to social desirability. Similarly, the use of self-reported GDM diagnoses may not be as accurate as those from medical records. However, previous studies have shown that self-reported GDM diagnoses are valid for research purposes [47].

Despite the limitations, the study has several strengths. First, to the best of our knowledge, this was the first study on the PA and SB of Asian immigrants in the US with prior GDM who are disproportionately affected by GDM [2] and also at high risk for T2D. Next, the use of accelerometers to collect objective PA and SB provided solid empirical data on their inactive and sedentary lifestyle pattern. In addition, 103 out of 106 participants had high adherence to the study protocol in this study which resulted in a high number of days with valid data. The successful recruitment (the recruitment target of 100 was met within a reasonable timeline even with the challenges of the pandemic) and high adherence to the study protocol indicated that WeChat is a promising low-cost platform for further intervention studies for this high-risk population. Lastly, we considered unique immigration-related and cultural factors for this understudied risk group at a unique life stage after childbirth, which will inform the development of a much-needed culturally sensitive intervention for this group to increase the relevance and effectiveness of the current intervention effort.

## 5. Conclusions

In conclusion, an inactive and sedentary lifestyle pattern was identified among Chinese immigrants with prior GDM. The development and evaluation of culturally sensitive PA interventions to decrease acculturative stress and improve family/social support in this population are warranted.

## Figures and Tables

**Table 1 ijerph-19-11409-t001:** Socio-demographic and clinical characteristics of Chinese immigrants in the US with prior GDM (n = 106).

Variables	Mean (SD)or n (%)
Age	34.3 (3.7)
Age of arrival (year)	25.0 (5.5)
<18 years old	7 (6.9%)
18–25 years old	50 (49.0%)
26–30 years old	29 (28.4%)
31–40 years old	16 (15.7%)
Length of stay (year)	9.2 (4.8)
≤3 years	9 (8.8%)
4–10 years	59 (57.8%)
11–30 years	34 (33.3%)
BMI (kg/m^2^)	21.7 (2.6)
BMI < 18.5	13 (12.3%)
BMI 18.5–23	67 (63.2%)
BMI ≥23	26 (24.5%)
Education	
Lower than college	8 (7.7%)
College degree	21 (20.2%)
Master or above	75 (72.1%)
Employment status	
Employed/students	70 (67.3%)
Unemployed	34 (32.7%)
Annual household income	
$80 K or less	31 (30.1%)
more than $80 K	72 (69.9%)
Immigration status	
Citizenship	16 (15.7%)
Permanent resident	54 (52.9%)
Temporary visa	32 (31.4%)
Language preference	
English	69 (65.1%)
Chinese	37 (34.9%)
Birthplace	
Mainland China	100 (96.2%)
Taiwan	4 (3.9%)
Family history of DM	
Yes	34 (32.1%)
No	72 (67.9%)
Number of children	
One	49 (46.2%)
Two	49 (46.2%)
Three or more	8 (7.5%)
Number of pregnancies with GDM	
One	78 (73.6%)
Two or more	28 (26.4%)
Currently breastfeeding	
Yes	39 (36.8%)
No	67 (63.2%)
History of postpartum depression	
Yes	6 (5.7%)
No	100 (94.3%)
Age of the youngest child	
Younger than 1	50 (48.5%)
1~2-year-old	25 (24.3%)
2~5-year-old	28 (27.2%)
Ever use of insulin as GDM treatment	
Yes	18 (17.0%)
No	88 (83.0%)
Charlson Comorbidity Index Score	
0	89 (85.6%)
1	9 (8.7%)
2–3	6 (5.8%)

**Table 2 ijerph-19-11409-t002:** Risk perception, knowledge, cultural and psychosocial characteristics of Chinese immigrants in the US with prior GDM (n = 106).

Constructs	Variables	Mean (SD) or n (%)	Possible Range
Knowledge & Risk perception	The overall perceived T2D risk	2.31 (0.95)	1–4
	Diabetes Risk Knowledge subscale	0.63 (0.14)	0–1
	Knowledge of PA guideline		-
	Yes	42 (40.8%)	
	No	61 (59.2%)	
Cultural factors	Ethnic society immersion	3.69 (0.35)	1–4
	Dominant society immersion	2.59 (0.60)	1–4
	Acceptance of Doing-the-month postpartum practice	2.74 (0.58)	1–5
	Were they currently living with their parents?		-
	Yes	19 (18.3%)	
	No	85 (81.7%)	
	Were they currently living with in-laws?		-
	Yes	8 (7.8%)	
	No	95 (92.2%)	
Psychosocial factors	Social support for PA from family	1.89 (1.26)	0–4
	Social support for PA from friends	0.93 (0.99)	0–4
	Social support for PA from family and friends	1.41 (0.87)	0–4
	Self-efficacy for PA	1.46 (0.94)	0–4
	Acculturative stress	1.83 (0.64)	1–5

**Table 3 ijerph-19-11409-t003:** Characteristics of the Actigraph data among Chinese immigrants in the US with prior GDM (n = 103).

Characteristics	Mean (SD)	Median [IQR]
Number of days with valid ActiGraph data	6.79 (0.52)	7 [7–7]
7 days	85 (82.5%)	
6 days	15 (14.6%)	
4–5 days	3 (2.9%)	
Average hours of wear time per day	14.23 (1.37)	14.23 [13.24–15.09]
Average hours of sedentary time per day	9.19 (1.44)	9.01 [8.11–10.31]
Prolonged sedentary time	2.49 (1.37)	2.45 [1.47–3.37]
Non-prolonged sedentary time	6.69 (1.10)	6.81 [6.03–7.47]
% in sedentary time among the total wear time	64.67% (7.84%)	65.26% [58.68%–70.67%]
Average minutes of light activity time per day	285.50 (71.02)	282.14 [233.83–344.71]
% in light activity time among total wear time	33.47% (7.69%)	33.32% [27.47%–39.74%]
Average minutes of moderate activity time per day	14.48 (12.70)	10.50 [5.28–20.28]
% in moderate activity time among total wear time	1.73% (1.57%)	1.21% [0.59%–2.39%]
Average minutes of vigorous activity time per day	1.14 (4.10)	0 [0–0.29]
% in vigorous activity time among total wear time	0.13% (0.49%)	0.0% [0.0%–0.03%]
Average minutes of MVPA time per week	109.23 (96.66)	79 [38–151]
% in MVPA time among total wear time	1.86% (1.70%)	1.28% [0.59%–2.49%]
Meeting PA guidelines or not	28 (27.2%)	
Number of days with 30 min MVPA or more	1.24 (1.85)	0 [0–2]
<1 day	55 (53.4%)	
1–2 days	28 (27.2%)	
3–4 days	10 (9.7%)	
>5 days	10 (9.7%)	
Number of days with MVPA < 30 min when the previous day was also MVPA < 30 min	4.15 (2.17)	5 [3–6]
% of days with MVPA < 30 min when the previous day was also MVPA < 30 min among all the possible days	72.30% (36.39%)	100% [50%–100%]
0%	12 (11.7%)	
1–50%	18 (17.5%)	
51–99%	18 (17.5%)	
100%	55 (53.4%)	
Average steps per day	5908.06 (2138.14)	5760.67 [3936.00–7364.00]

**Table 4 ijerph-19-11409-t004:** Adjusted effects on MVPA Minutes Per Week in the final model (n = 103).

	B	Standardized B	SE of B	*p*
BMI	0.061	0.160	0.034	0.074
Annual household income $80 K or less	−0.340	−0.154	0.200	0.092
Knowledge of PA guidelines	0.316	0.154	0.179	0.081
Currently NOT living with parents	−0.477	−0.184	0.230	0.041
Acculturative stress	−0.471	−0.293	0.143	0.001
Social support for PA	0.180	0.155	0.104	0.085

Note: The outcome variable MVPA minutes per week was log-transformed. R^2^ = 0.31 in this model.

**Table 5 ijerph-19-11409-t005:** Adjusted effects on Meeting PA Guidelines in the final model (N = 103).

	B (SE)	OR (95% CI)	*p*
Annual household income $80 K or less	−1.195 (0.665)	0.303 (0.082–1.114)	0.072
Permanent resident (vs. citizens)	−1.866 (0.743)	0.155 (0.036–0.664)	0.012
Age of youngest child ≥ 1 (vs. <1)	−1.410 (0.576)	0.244 (0.079–0.755)	0.014
Currently NOT living with parents	−1.715 (0.616)	0.180 (0.054–0.602)	0.005

**Table 6 ijerph-19-11409-t006:** Adjusted effects on Steps per Day in the final model.

	B	Standardized B	SE of B	*p*
Currently NOT living with parents	−1682.404	−0.306	522.760	0.002

Note: R^2^ = 0.09 in this model.

**Table 7 ijerph-19-11409-t007:** Adjusted effects on Prolonged Sedentary Time in the final model.

	B	Standardized B	SE of B	*p*
Employed (vs. unemployed)	1.213	0.414	0.263	0.000
Having one child (vs. two or more child)	0.461	0.168	0.248	0.065

Note: R^2^ = 0.22 in this model.

**Table 8 ijerph-19-11409-t008:** Adjusted effects on Percentage of SB time in the final model.

	B	Standardized B	SE of B	*p*
BMI	−0.007	−0.238	0.003	0.006
Employed (vs. unemployed)	0.077	0.463	0.014	0.000

Note: R^2^ = 0.29 in this model.

## Data Availability

Data available on request due to restrictions (e.g., privacy). The data presented in this study are available on request from the corresponding author.

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
