# Peer review of "Objectively-Measured Physical Activity and Sedentary Behaviors and Related Factors in Chinese Immigrants in the US with Prior Gestational Diabetes Mellitus"

_ijerph, 2022, doi:10.3390/ijerph191811409_

Round 1
Reviewer 1 Report
This survey examined factors associated with exercise and life activity in the limited population which met two conditions, as well as their trends in the population. It may be of value for the people in the population, but may be a little for the other people. The manuscript does not describe enough about how this survey could be useful to the general public.
Major Comments:
#1 What information people with prior GDM who are not immigrants and/or immigrants without prior GDM can get from this survey? Or, can’t they do them?
#2 How do you rate the impact of having a GDM history and being an immigrant, respectively, on low PA level and high SB time? If there were not any data to evaluate it, why did you just examine this unique population at first before examining at each condition?
#3 How do you evaluate the bias in the survey subjects recruited at WeChat? What is the basis for judging that the recruitment via the SNS was the “successful” recruitment (L413)?
Minor comments
#1 L325 : the youngest child >= 10 should be “>=1”.
#2 LL337-338: Numbers to be compared should have the same number of digits after the decimal point.
#3 LL375-376: The sentence after HENCE should modified or deleted to avoid over-discussion.
Reviewer 2 Report
Huang et al., investigated the physical activity and sedentary behaviors in a cohort Chinese immigrants in the USA with prior gestational diabetes with the aim of providing a starting point/framework in developing future culturally sensitive interventions. I found the paper succinct, well-written with a balanced discussion on the positives/limitations of the study especially during the COVID-19 pandemic. I would like the authors to offer some guidance in utilizing this data for future interventions. For instance, if post partum women have maintained motivation to lead a healthier lifestyle close to GDM experience, what are the interventions available (if any) to maintain the lifestyle for longer.
Reviewer 3 Report
Please further clarify why conclusions are drawn from by model 5 at 0.1 level. High probability for error may exist at 0.1. Limitations of study should be more clearly defined, e.g self‐reporting of GDM diagnosis may not be accurate.
Round 2
Reviewer 1 Report
In the revised version, all doubts have been resolved. Now I consider it worthy of acceptance.